# Bioinformatics and Expression Analysis of the Chitinase Genes in Strawberry (*Fragaria vesca*) and Functional Study of *FvChi-14*

**DOI:** 10.3390/plants12071543

**Published:** 2023-04-03

**Authors:** Tiannan He, Jianshuai Fan, Gaozhen Jiao, Yuhan Liu, Qimeng Zhang, Ning Luo, Bilal Ahmad, Qingxi Chen, Zhifeng Wen

**Affiliations:** 1College of Horticulture, Fujian Agriculture and Forestry University, Fuzhou 350002, China; 2Agricultural Genomics Institute at Shenzhen, Chinese Academy of Agricultural Sciences, Shenzhen 518120, China

**Keywords:** biotic stress, pathogen, genome, evolutionary history, PR3

## Abstract

Plant chitinases (EC 3.2.1.14) are pathogenesis-related (PR) proteins and are well studied in many plant species. However, little is known about the genomic organization and expression of chitinase genes in strawberries (*Fragaria vesca*). Here, 23 *FvChi* genes were identified in the genome of strawberry (*F. vesca*) and divided into GH18 and GH19 subfamilies based on phylogenetic relationships. A detailed bioinformatics analysis of the *FvChi* genes was performed, including gene physicochemical properties, chromosomal location, exon–intron distribution, domain arrangement, and subcellular localization. Twenty-two *FvChi* genes showed upregulation after *Colletotrichum gloeosporioides* infection. Following the exogenous application of SA, *FvChi-3*, *4*, and *5* showed significant changes in expression. The ectopic expression of *FvChi-14* in *Arabidopsis thaliana* increased resistance to *C. higginsianum* via controlling the SA and JA signaling pathway genes (*AtPR1*, *AtICS1*, *AtPDF1.2*, and *AtLOX3*). The FvChi-14 protein location was predicted in the cell wall or extracellular matrix. We speculate that *FvChi-14* is involved in disease resistance by regulating the SA and JA signaling pathways. The findings of this study provide a theoretical reference for the functional studies of *FvChi* genes and new candidates for strawberry stress resistance breeding programs.

## 1. Introduction

Chitinases (EC.3.2.1.14) are glycosyl hydrolases widely found in various organisms in nature, and are thought to be one group of pathogenesis-related (PR) proteins [1]. Normally, these proteins are expressed at basal levels, but pathogen attacks or abiotic stress can significantly increase their expression, leading to systemic acquired resistance (SAR) in plants [2]. Chitinase mainly hydrolyzes β-1,4 glycosidic bonds in chitin polymers to generate N-acetyl-D-glucosamine (GlcNAc) oligomers, which interfere with or degrade chitin [3]. Chitin is a major structural component of fungal cell walls, and thus chitinase plays critical roles in plant defense against pathogens [4]. Viral/fungal infections induce plant chitinase activity, through which plants receive signals about the fungal attack. Apoplastic chitinase induces the release of fungal inducers, and vacuolar chitinase then acts to defend against pathogens [5]. Apoplastic chitinase produces disease-related signals that induce plant immunity, and vacuolar chitinase degrades newly synthesized chitin chains and inhibits fungal growth [6]. Plant chitinases can be divided into two subfamilies: 18 (GH18) and 19 (GH19) of glycosyl hydrolases, which can be further divided into five distinct classes (class I–V). Classes I, II, and IV are members of family GH19, while classes III and V belong to family GH18 [7]. Chitinases of family GH18 are ubiquitous among organisms, whereas family GH19 chitinases are restricted to plants and *Streptomyces* [8]. Members of the GH19 subfamily have an N-terminal chitin-binding domain (CBD) structure that can combine with chitin to enhance plant defense against pests and stress resistance [3].

The antifungal effects of chitinase in plants have been demonstrated in many studies. Chitinase genes have critical roles in multiple processes, including plant development and defense against stress. For example, after inoculating *MnChi18* transgenic *Arabidopsis* with *Botrytis cinerea*, the activity of catalase (CAT) and the amount of malondialdehyde (MDA) decreased, showing that overexpression of *MnChi18* could boost the resistance to *B. cinerea* [9]. The transient expression of *CaChiIII7* significantly induced the expression of many defense-related genes and the high expression in pepper leaves along with H_2_O_2_ biosynthesis, and improved the basic resistance of pepper to *C. acutatum* [10]. When the resistant variety of cucumber ‘JIN5-508′ was inoculated with *Podosphaera xanthii*, the expression level of the chitinase genes increased significantly, which triggered a papillary reaction to stop the invasion of powdery mildew [11]. Studies on the molecular control of defense responses in pine trees have revealed that after being attacked by potential signaling molecules found in angiosperms and necrotizing diseases, plants produce several chitinase homologs [12]. Chitinase genes *Chi23*, *Chi32*, and *Chi47* were silenced, and this significantly decreased cotton’s resistance to *Verticillium dahliae*, indicating their function in *V. dahlia* resistance [13]. Moreover, some studies have reported the role of chitinase genes in abiotic stress tolerance. The expression of the class III *AtChiA* gene was upregulated following salt and wound stresses [14]. *ScChiI1* and *ScChiIV1* showed upregulation under PEG, NaCl, and CuCl_2_ stress [15]. Strawberry (*Fragaria ananassa* Duch.) is a high-value fruit popular among the public for its delicious taste and rich nutrition [16]. However, strawberry fruit is vulnerable to mechanical damage and infection by plant pathogens, including viruses, bacteria, and fungi, during harvesting and storage due to the lack of shell protection [17]. Anthracnose is a serious fungal disease caused by the necrotizing fungus *Colletotrichum* spp., which mostly infects strawberry roots and stolons during development [18]. The main anthracnose species reported to infect strawberries are *C. gloeosporioides* and *C. fragariae* in China [19]. One of the most prevalent types of strawberry PR genes with hydrolytic activity has been found in the chitinase family [20]. There is little known about the roles of chitinase genes in antifungal disease resistance in strawberries [17,21,22]. Detailed information on the disease resistance mechanisms and pathways of chitinase genes in strawberries is lacking. 

The chitinase gene family has been studied in a variety of plants, such as cotton (*Gossypium*) [13], cucumber (*Cucumis sativus*) [23], tomato (*Solanum lycopersicum*) [24], tea (*Camellia sinensis*) [5], cabbage (*Brassica oleracea*) [25], mulberry (*Morus notabilis*) [9], domesticated apple (*Malus domestica*), and wild apple (*Malus sieversii*) [26]. However, less information is available about the roles of chitinase genes in strawberry. The important roles of chitinase genes in different crops and lack of detailed information in strawberry justifies the need of identification, expression profiling, and functional characterization of chitinase genes in this crop. We executed detailed bioinformatics analysis of chitinase genes in *F. vesca*, including determining gene numbers, gene structures, motif distribution patterns, and evolutionary history analysis. Further, we performed expression profiling of chitinase genes in different plant parts and following *C. gloeosporioides* infection and hormone treatments. Based on the results of the initial study, *FvChi-14* was isolated from strawberry and overexpressed in *Arabidopsis* for functional studies. Overexpression of *FvChi-14* significantly increased the resistance of transgenic *Arabidopsis* to anthracnose. Our study provides new candidates for disease resistance breeding of strawberry and insights about the functions of *FvChi.*

## 2. Results

### 2.1. Identification and Physiochemical Properties of FvChi Proteins

Twenty-three annotated *FvChi* members with chitin hydrolysis or chitinase-like functions were found in the *F. vesca* genomes (Table 1). The physicochemical properties and basic information are mentioned in Table 1. Sequence analysis showed that the amino acids of FvChi members ranged in length from 227 AA (FvChi-23) to 777 AA (FvChi-3), with protein molecular weights of 24.662 (FvChi-23) to 87.535 (FvChi-3) kDa and isoelectric points (pI) of 4.59 (FvChi-23) to 8.95 (FvChi-6). *FvChi-1* to *FvChi-22* were unevenly distributed on six chromosomes, and there was no member on chromosome four. Interestingly, *FvChi-23* was present on an unknown chromosome (Figure 1). The highest and the lowest number of genes were present on chromosomes 1 (9 genes) and 5 (1 gene), respectively. Most of the FvChi members were predicted to be localized in the cell wall (39.1%) and vacuole (39.1%). 

### 2.2. Phylogenetic Studies of FvChi Genes

Based on the phylogenetic tree constructed among chitinase genes of strawberry, rice, and *Arabidopsis* (Figure 2), *FvChi* genes were divided into two subfamilies, GH18 and GH19. There were 17 and 6 genes in the GH18 and GH19 subfamilies, respectively. Moreover, GH18 and GH19 can be divided into subclasses (categories) having two and three classes, respectively (Figure 3A). According to motif and domain analysis, there were close relationships among members of the same subfamily (Figure 3B,C and Appendix A). However, differences were found in the identified motifs of the two subfamilies GH18 and GH19. For example, three members (*FvChi-3*, *FvChi-5* and *FvChi-8*) of class V were highly similar, and all have a catalytic domain of serine/threonine kinases (STK). Two members (*FvChi-2* and *FvChi-15*) of class IV have a lysozyme-like domain whose family contains several members, including chitinase. Furthermore, two members (*FvChi-7* and *FvChi-11*) of class I have a chitin-bind domain (CBD), which has a role in the recognition and binding of chitin subunits. The members of class III and class V apparently have a distinct glycosyl hydrolase family 18 (GH18) domain. Motif 8 was found in all members of the GH18 subfamily, indicating its importance for the strawberry chitinase GH18 subfamily, while motifs 7 and 16 were only found in GH19 subfamily proteins. However, motifs 9, 14, and 15 were found in both subfamilies.

### 2.3. Gene Structure Architecture of FvChi Genes and Expansion Patterns

Exon–intron analysis revealed significant differences in gene structure and number of introns between the GH18 and GH19 subfamilies (Figure 4A). A total of 64.7% members of the GH18 subfamily do not have introns, while all members of the GH19 subfamily have introns. Further, members from the same class exhibited a closely linked gene structure based on a similar number or length of exons. Three genes (*FvChi-3*, *FvChi-5*, and *FvChi-8*) of class V have the most introns (six), while other members have one or no intron. Intraspecies collinearity analysis was executed to elaborate the evolution and origin dynamics of *FvChi* genes (Figure 4B). Three groups of genes with high homology showed clear genetic relationships, revealing that they evolved by tandem replication and formed gene clusters with similar sequences and functions. Only one set of collinearity of segmentally duplicated genes (*FvChi-19*, *20*, *21*, and *FvChi-22*) was identified on chromosome 7.

### 2.4. Analysis of Cis-Acting Elements in FvChi Gene Promoters

The *cis*-elements contained in the promoter region of the *FvChi* member have various roles in plant growth and development. *Cis*-elements were mainly sensitive to abscisic acid (ABA), salicylic acid (SA), methyl jasmonate (MeJA), anaerobic, and drought conditions (Figure 5 and Appendix A). MeJA-sensitive elements were identified in the promoter region of the 18 *FvChi* members, and the highest numbers were present in *FvChi-3*, *FvChi-5*, and *FvChi-8*. The promoters of nine genes contained SA-responsive elements, suggesting that most *FvChi* members might have roles in biotic stress tolerance. Abscisic acid-responsive elements (ABREs), ARE (anaerobic induction), and LTR (low-temperature) elements were identified in the promoter regions of 17, 17, and 18 genes, respectively. Elements associated with anoxia stress were found in 18 *FvChi* genes. Moreover, TC-rich repeats, ACE (light), MBS (drought), AuxRE (auxin), circadian, CAT-box (meristem), and GARE (gibberellin)-responsive *cis*-elements were also found in the promoter regions of some members. 

### 2.5. Expression Analysis of FvChi Gene in Tissue/Organs

Expression patterns of *FvChi* members were analyzed by quantitative reverse transcription PCR (qRT-PCR) (Figure 6A). Compared to other tissues/organs, most of the genes were highly expressed in roots such as *FvChi-1*, *2*, *4*, *6*, *9*, *10*, *11*, *13*, and *18*. On the other hand, the expression levels of *FvChi* members in flowers were very low. *FvChi-15* and *19* showed stem specific expression and *FvChi* genes displayed ubiquitous expression in leaves. *FvChi-12*, *14*, *16*, *17*, *20*, and *22* showed high expression in fruits. As anticipated, closely related genes showed similar expression patterns; for example, members of class III (*FvChi-12*, *16*, and *22*) showed high expression in fruits and low expression in flowers. The results suggested that the *FvChi* genes have different roles in different strawberry organs, and homologs may have similar functions.

### 2.6. Expression Analysis of FvChi Genes Following C. gloeosporioides Inoculation

The expression levels of *FvChi* genes were measured in strawberry leaves following *C. gloeosporioides* inoculation (Figure 6B). *FvChi* genes exhibited different expression patterns and can be divided into four groups based on expression levels. The Group I (*FvChi-19*) gene was downregulated over time after inoculation with *C. gloeosporioides*. The Group II (*FvChi-8*) gene initially showed upregulation and then was subsequently downregulated after 6 h. The members (*FvChi-1*, *2*, *3*, *4*, *5*, *6*, *7*, *9*, *10*, *11*, *12*, *13*, *14*, *15*, *16*, *17*, *18*, *20*, *21* and *22*) of Group III also showed the same trend—initially upregulation and subsequent downregulation—but showed a peak expression at 12 hpi. Group IV contained only *FvChi-23*, which is upregulated at 24 hpi and subsequently downregulated. Overall, the relative expression levels of most *FvChi* genes increased significantly after *C. gloeosporioides* infection and reached a maximum at 12 hpi. 

### 2.7. Expression Analysis of Selected FvChi Genes Following SA and MeJA Treatments

The responses of six *FvChi* genes (*FvChi-3*, *4*, *14*, *15*, *22* and *23*) were estimated against two hormone (SA and MeJA) applications (Figure 6C). Following SA treatment, *FvChi-3*, *14*, and *15* showed the highest increase (23.00, 36.67, and 27.47, respectively), followed by a subsequent decrease. The other three genes showed a slight change in expression at 6 hpt. Overall, most of the genes showed peak expression at 6 hpt. As far as the response to MeJA application is concerned, there was no significant change in expression.

### 2.8. FvChi14 Overexpression in A. thaliana Enhances Resistance to C. higginsianum

Bioinformatics and expression profiling revealed that the *FvChi14* gene was abundantly induced in strawberry when inoculated with *C. gloeosporioides*, and the expression was maximally upregulated with external hormone treatment (SA). Therefore, the *FvChi14* gene was selected for more detailed analysis and overexpressed in *Arabidopsis* (Appendix A). A total of 43 T2 transgenic plants were obtained, and out of these, 3 independent T3 transgenic lines (T3-1, T3-2, and T3-26) having the strongest resistance to *C. higginsianum* were chosen for further studies. The *FvChi-14* transcript was not shown in the WT, and the highest expression was noticed in the transgenic line T3-2 and the lowest expression in T3-26 (Figure 7G). After 48h of *C. higginsianum* inoculation with suspended drops, the leaves of the three transgenic lines and Col-0 plant showed obvious translucent necrotic spots surrounded by a yellow halo. However, the lesion diameter of the transgenic plant leaves was significantly smaller than that of the wild type (Figure 7A,B). To find the molecular basis of *FvChi-14* resistance to *C. higginsianum*, the expression levels of some genes having presumed roles in SA and JA signaling pathways were detected using qRT-PCR (Figure 7C–F). *PR1* is a SAR molecular marker downstream of SA accumulation; *ICS1* is required for SA biosynthesis. At 12 hpi, *AtPR1* showed 37.5- and 22.1-fold higher expression in the T3-1 and T3-26 transgenic lines, respectively, compared to WT plants (Figure 7C). *AtPR1* was upregulated in T3-26 at 48 hpi but downregulated in T3-1 and T3-2 (Figure 7C). The relative expression of *AtICS1* in transgenic strains showed a similar trend and was significantly higher than that of WT at 24–48 hpi (Figure 7D). In particular, the expressions of *AtICS1* in T3-2 and T3-26 strains were 3.00- and 4.30-fold higher than that of WT at 24 hpi, respectively. *AtICS1* was consistently upregulated in T3-1 at 48 hpi, while the other transgenic strains started to be downregulated. *AtPDF1.2* and *AtLOX3* have roles in JA signaling pathways. *AtPDF1.2* showed higher expression levels at all time points in transgenic lines compared to WT. *AtPDF1.2* in T3-1 was significantly upregulated between 24 and 48 hpi, while it showed slight downregulation in T3-2 and T3-26. Moreover, *AtPDF1.2* was overexpressed 20.5-fold higher in T3-1 compared to WT at 48 hpi (Figure 7E). Similarly, the relative expression of *AtLOX3* was 3.10-fold higher in T3-26 than in WT at 48 hpi (Figure 7F). The relative expression of *AtLOX3* peaked at 12 hpi in T3-1 and 2, and at 24 hpi in T3-26. The expression patterns of defense-related genes were different in each transgenic line.

### 2.9. Subcellular Localization of FvChi-14 in Onion Cells

The subcellular location of the FvChi-14 was validated by transient expression, using GFP protein as a tag in onion cells. Onion epidermal cells were transfected with *Agrobacterium* containing 35S:FvChi-14:GFP fusion vector and 35S:GFP was used as control. Confocal microscopic examination of onion epidermal cells revealed that the FvChi-14 protein was predicted to be localized to the cell wall or extracellular matrix and exhibited green fluorescence signals (Figure 8).

### 2.10. Prokaryotic Expression and Western Blot Analysis of FvChi-14

To investigate the potential resistance to abiotic stress, FvChi-14 was ectopically expressed in *E. coli* BL21 (DE3). The recombinant strain pGEX-4T-1-FvChi-14 showed a specific band at about 30kD in SDS-PAGE analysis after inducing protein expression (Figure 9A), which was similar to the protein size predicted by the ProtParam tool. Western blot analysis further verified that the specific band appeared at the corresponding position of the recombinant protein (Figure 9B), indicating that FvChi-14 protein was correctly expressed in *E. coli*.

## 3. Discussion

In response to developmental cues and pathogen attacks, higher plants have evolved several defense mechanisms. Different pathogenesis-related (PR) protein-coding genes are expressed in plants [27]. PR3/chitinases are the most important PRs, playing vital roles in resistance to pathogen infection. PR3 are lytic enzymes found in many higher plants and can directly attack the structural components of fungi and insects [28]. Nowadays, anthracnose caused by *Colletotrichum* spp. has seriously affected the development of the strawberry industry [29]. Chitinases are considered one of the key factors in plant resistance to fungal diseases. The transformation of chitinase genes in resistant strawberry varieties via transgenic technology has attracted the attention of researchers. Therefore, we conducted a detailed and systematic study to identify and characterize the chitinase gene family in the strawberry genome. Twenty-three chitinase genes were found in the diploid strawberry genome, and one gene (*FvChi-14*) was selected for functional studies. There are 24 chitinase genes in *A. thaliana* [7]. The close number of genes suggest that these species might have had a close relationship before speciation. However, the other studied species have different numbers of genes, for example, *Brassica juncea***,**
*Capsicum annuum*, and *Glycine max* have 47, 31, and 38, genes, respectively [2,3,30]. The different numbers of chitinase genes may be due to the differences in ploidy level and genome size between species [31].

The phylogenetic studies of chitinase sequences in *F. vesca*, *A. thaliana*, and *O. sativa* indicate that the proportion of class III genes in strawberries and rice is high (Figure 2). This may be related to the fact that class III chitinases are relatively conserved in different crops. The lysozyme activity of some class III chitinases suggests that plant class III chitinases may have biological effects different from those of other classes [32]. It has been shown that different class III chitinase isomers respond differently to *Fusarium oxysporum* infection, and melon resistance to *F. oxysporum* infection is linked with some subtypes of this enzyme [33]. *Oschib1* and *Oschib2* acted with extracellular chitinase to hydrolyze chitin oligosaccharides released from the cell wall of the pathogenic fungi [34]. In our experiment, the expression levels of the class III chitinase genes, *FvChi-9* and *FvChi-18*, were upregulated by 55.70 and 54.20 times, respectively, after inoculation with *C. gloeosporioides*. Future research is needed to explore the roles of class III chitinases in disease resistance. 

Gene intron density and the regulation of genes in response to stress are inversely correlated [35]. Because long and abundant introns extend transcription time, genes with few introns can be rapidly regulated following stress [30]. Stress-related gene families such as *FvbZIP* [36], *SsLRR-RLK* [37], and *TaZF-HD* [38] all have lower numbers of introns. Our results are also in line with these findings, and among the 23 chitinase genes in *F. vesca*, 11 genes did not contain introns and 9 genes had 1 to 2 introns (Figure 4A). Based on the previous findings and our results, we surmise that most of the *FvChi* genes can have roles in disease resistance. Moreover, the occurrence of stress-related *cis*-elements (Figure 5), changes in the expression of genes following *C. gloeosporioide* inoculation, and the response to SA treatments also support this hypothesis. However, functional analysis is required for confirmation.

Based on the bioinformatics and response against *C. gloeosporioides*, SA, and MeJA treatments, the *FvChi-14* gene belonging to class II was selected for functional studies (Figure 6). Our findings are in line with earlier research showing that class II chitinases can be activated by ethylene, salicylic acid, or fungal spore treatment, and are extensively expressed in healthy plants [39]. Class II chitinases may perform various roles such as playing a role in plant defense and endogenous regulatory functions during plant development [40]. Overexpression of *FvChi-14* in *A. thaliana* increased resistance to the fungus *C. higginsianum*. Following in vitro leaf inoculation with *C. higginsianum*, the lesion size on the leaves of transgenic lines (T3-1, T3-2 and T3-26) was significantly smaller than that of the wild plants. After this, the whole plant was sprayed with inoculated pathogen suspensions, and the relative expression levels of *AtPR1*, *AtICS1*, *AtPDF1.2*, and *AtLOX3* were measured at 0, 3, 6, 12, 24, and 48 hpi. These resistance-related genes showed higher relative expression levels at 12-48 hpi in *FvChi-14* transgenic plants than in control plants (Figure 7). 

Our results are supported by previous findings; for example, overexpression of barley class II chitinase genes in transgenic wheat enhanced resistance to *Fusarium graminearum* in greenhouse and field conditions [41]. The Na_2_CO_3_-responsive chitinase gene *LcCHI2* of class II improved pathogen resistance in transgenic maize and tobacco [42]. Compared to control cultivars, transgenic sugarcane lines treated with the barley chitinase class II gene exhibited better resistance to inoculation with *C. falcatum* [43]. Furthermore, the relative expression of *AtPR1* peaked earlier at about 12hpi; *AtPR1* expression levels were upregulated in T3-26 and downregulated in T3-1 and T3-2 at 48 hpi, while the relative expression of *AtPDF1.2* peaked later at 48hpi, especially in the T3-1 line. This may be related to the fact that the genus *Colletotrichum* spp. followed the hemibiotrophic mode of infection [44]. For defense against biotrophic pathogens, SA is a main signaling molecule, while JA-mediated defense signaling is usually effective against necrotizing pathogens [45,46]. A potential explanation to these results could be as follows: *FvChi-14* overexpression lines in *A. thaliana* first upregulate the relative expression level of the SA pathway-related gene *AtPR1* and then upregulate the relative expression level of the JA defense pathway gene *AtPDF1.2* after *C. higginsianum* infection. In conclusion, overexpression of the *FvChi-14* gene can improve the resistance of *A. thaliana* to *C. higginsianum*. SA is a natural signaling molecule that activates the expression of defense genes and triggers changes in components of the redox signal transduction pathway [47]. SA scavenges hydroxyl radicals in plants and prevents the inactivation of catalase by H_2_O_2_, thus protecting plants from oxidative damage [48]. Chitinase genes are a class of defense response genes that are closely related to SAR in plants. SAR is often accompanied by the production of reactive oxygen species (ROS). Chitinase may induce the SA pathway by increasing ROS in response to disease resistance in plants. Chitinase has been shown to be induced by JA rather than SA, but the two pathways are not completely independent and cross each other at different sites [49]. Studies have shown that ROS may be involved in JA-induced chitinase accumulation in leaves and leaf sheaths in rice [50]. External application of MeJA enhanced the activity of inductive-type chitinase [51]. However, in this study, MeJA did not significantly induce the expression of some strawberry chitinase genes. (Figure 6C). Induction of SA and JA signaling patterns can alter gene expression and lead to specific defense responses to stress in plants [10]. The specific action mechanism of chitinase in these signaling modes needs further study. As anticipated, most chitinase proteins were predicted in the cell wall and vacuole in strawberry (Table 1). An N- or C-terminal targeting region on all plant chitinases sends the enzymes first to the endoplasmic reticulum and subsequently to either the vacuole or the apoplast [52]. Moreover, FvChi-14 protein was predicted to act in the extracellular matrix or vacuole, and the transient expression of tag protein GFP in onion epidermal cells confirmed that the subcellular location of FvChi-14 is the cell wall or the extracellular matrix (Figure 8).

## 4. Materials and Methods

### 4.1. Plant Materials and Seedlings Treatment

The strawberry (*F. vesca* ssp.) and *A. thaliana* (Col-0) plants grown in the plant incubators were used as experimental material. The growing conditions for strawberry were maintained at 23–25 °C and 85% humidity, while for *A. thaliana*, conditions were 21 °C, 70% relative humidity, and a 12 h light/12 h dark cycle. 

Pathogen inoculation (*C. gloeosporioides* and *C. higginsianum*) and plant maintenance were carried out according to our previous experiment [53]. For the purpose of determining the biological efficacy of *FvChi-14* against anthracnose, *C. higginsianum* was used to infect both *A. thaliana* transgenic lines and the Col-0 wild types [54]. For RNA extraction, leaf samples were collected at 0, 6, 12, 24, and 48 h postinoculation (hpi). Strawberry leaves were treated with 1.5 mL of either 50 mM methyl jasmonate (MeJA) or 5 mM salicylic acid (SA) solutions, and distilled water was used as a control. After spraying, the leaves were collected at 0, 3, 6, 12, and 24 h post-treatment (hpt). Strawberry leaves, stolons, flowers, fruits, and roots were collected and quickly immersed in liquid nitrogen for constitutive expression analysis. After this, samples were stored at −80 °C for further use and every experiment was performed with three biological and technical replicates.

### 4.2. Identification of Chitinase Genes in Strawberry

The genomes and gene annotation files of *F. vesca* were retrieved from NCBI (https://www.ncbi.nlm.nih.gov/genome/browse#!/overview/, accessed on 10 January 2021). The chitinase protein sequences of *A. thaliana* and *O. sativa* were downloaded from the Phytozome database (https://phytozome.jgi.doe.gov/pz/portal.html, accessed on 20 January 2021). CDS was extracted from strawberry genomes by TBtools and translated into protein sequences. *A. thaliana* chitinase protein sequences were used as queries against *F. vesca* protein sequences. Possible members were obtained after blast comparison. The obtained sequences of chitinase proteins were verified using Swiss-Prot database and Batch CD-Search (https://www.ncbi.nlm.nih.gov/Structure/bwrpsb/bwrpsb.cgi, accessed on 30 January 2021). Twenty-three putative chitinase genes were screened and renamed *FvChi-1*- *FvChi-23* based on their position on the chromosome. Basic information about selected proteins, such as the start and end sites, was retrieved from the NCBI. 

### 4.3. Characterization Analysis of Strawberry Chitinase Genes

The accession number, chromosomal location, CDS length, and other information of *FvChi* genes were retrieved by NCBI. The physicochemical properties of *FvChi* genes were estimated using The ProtParam tool of ExPASy Server (https://web.expasy.org/protparam/, accessed on 10 February 2021), including protein molecular weight (MW) and isoelectric point (pI) values. The Plant-mPLoc Server (http://www.csbio.sjtu.edu.cn/bioinf/plant-multi/, accessed on 10 February 2021) was used for the prediction of subcellular locations of proteins.

### 4.4. Gene Structure and Motif Distribution Patterns Analysis

The members of the chitinase gene family were examined using the online MEME application (http://meme-suite.org/, accessed on 20 February 2021). All the default parameters were used, and the motifs were searched up to 20. The conserved domain of the members of the chitinase gene family was examined using the web application Batch CD-Search. TBtools was used to visualize the results [18], and Splign (https://www.ncbi.nlm.nih.gov/sutils/splign/splign.cgi?textpage=online&level=form, accessed on 20 February 2021) was used to study the intron and exon distribution patterns. The results were displayed using Gene Structure Display Server 2.0 (gao-lab.org).

### 4.5. Phylogenetic Studies of FvChi Genes

Protein sequences were used for phylogenetic tree construction. The phylogenetic trees were generated using MEGA X and the maximum-likelihood (ML) method with the following conditions: 1000 bootstrap values, the WAG model, and 95% partial deletion. The phylogeny was determined among FvChi proteins, and another was generated among chitinase proteins of *A. thaliana*, *O. sativa*, and *F. vesca*.

### 4.6. Expansion Pattern and Promoter Analysis of FvChi Genes 

The strawberry genomic database was compared with itself using the blast function of TBtools. The output results were imported into MCScan (http://chibba.pgml.uga.edu/mcscan2/, accessed on 22 February 2021) to determine collinear blocks under a default criterion [55]. The outcomes were shown using TBtools. The 2 Kb upstream nucleotide sequences of *FvChi* genes were downloaded and studied by PlantCARE (http://bioinformatics.psb.ugent.be/webtools/plantcare/html/, accessed on 22 February 2021) for *cis*-elements.

### 4.7. Expression Analysis of FvChi Genes 

Each sample was measured at 0.1 g, and total RNA was extracted from strawberry organs or treated leaves using RNAprep Pure Plant Kit (Tiangen, China). Sample total RNA concentration was measured using a Thermo Scientific NanoDrop 2000 instrument. Total RNA concentration was greater than 100 μg/μL, and OD260/280 values were 1.8–2.1. The first-strand cDNA was synthesized using the PrimeScript™RT kit (Takara) and contained 500ng total RNA per 10 μL cDNA reversal system. DNAMAN Version 10 software (https://www.lynnon.com/index.html, accessed on 27 February 2021) was used for designing primers, and Appendix A contains a list of the primers. The qRT-PCR was performed on Roche LightCycler96 machine for expression analysis. The PCR conditions were 30 s at 95 °C, 40 cycles of 5 s at 95 °C, 30 s at 60 °C, and a final step of 15 s at 72 °C. Each reaction mixture (20 μL) consisted of 0.5 μL of each primer (final concentration: 0.25 μM), 2 μL of diluted cDNA, 10 μL of TB Green *Premix Ex Taq*, 0.4 μL of ROX Reference Dye, and 6.6 μL of sterile water. The 2^−∆∆CT^ method was used for the calculation of relative expression values [56], and the *actin* gene (GenBank accession number AB116565) was used as an internal reference. Each reaction was repeated three times biologically and technically. 

### 4.8. Plasmid Construction and A. thaliana Transformation

Reverse transcription was used to create first strand of cDNA from 1 g of total RNA using the PrimerScriptTM-II 1st Strand cDNA Synthesis kit (TaKaRa Bio Inc. Dalian, China). The ORF sequence of the *FvChi-14* (972bp) was amplified with LA Taq (TaKaRa Bio. Inc.). The PCR product was subsequently cloned into pMD20-T (TaKaRa Bio Inc.) and sequenced (FuZhou ShangYa BioInc., Fuzhou, China). The resulting sequence was cloned into the binary vector pCAMBIA1300-HA (CAMBIA company) using specific primers *FvChi-14*-clone-F and *FvChi-14*-clone-R containing *BamH* I and *Spe* I sites (Appendix A). The sequence was also submitted to GenBank (Acc. NO. OQ211094). By using the freeze–thaw technique, the recombinant plasmid was transformed into *Agrobacterium tumefaciens* GV3101 [57], and then the recombinant was transferred into wild *A. thaliana* by floral dip method [58]. The solid MS medium containing 50 mg/L hygromycin was used for the screening of transgenic plants [59]. 

### 4.9. Subcellular Localization of FvChi-14 in Onion Cells

The ORF region of *FvChi-14* was cloned into the pGFPc vector using specific primers *FvChi-14*-PE-F and *FvChi-14*-PE-R containing *BamH* I and *X-bal* sites (Appendix A). The constructed fusion vector was verified by sequencing and transformed into *A. tumefaciens* GV3101 and was injected into onion epidermal cells for 48 h culture. The fluorescence reaction of GFP in the onion epidermis was observed by laser scanning confocal microscope (OLYMPUS IX83-FV3000).

### 4.10. Prokaryotic Expression and Western Blot Analysis of FvChi-14

The ORF region of *FvChi-14* was cloned into the pGEX-4T-1 vector using specific primers *FvChi-14*-PE-F and *FvChi-14*-PE-R containing *BamH* I and *Sma* I sites (Appendix A). The sequence of recombinant vector was verified by sequencing. The recombinant plasmid and original vector were introduced into *E. coli* BL21 (DE3) and cultivated in LB medium containing 100 mg·L^−1^ ampicillin.

The control strain (BL21: pGEX-4T-1) and recombinant strain (BL21: pGEX-4T-1- FvChi-14) were cultured at 200 rpm and 37 °C overnight. A total of 1% of the cultures were then inoculated to fresh LB medium containing 100 mg·L^−1^ ampicillin, and value was adjusted to 0.5 OD_600_. The recombinant strain was added with 0.2 mmol·L^−1^ isopropyl β-D-1-thiogalactopyranoside (IPTG) and cultured at 37 °C for 4 h to induce FvChi-14 protein expression. A total of 2 mL of the induced expression solution was centrifuged at 10,000× *g* for 1 min, and the bacteria were resuspended with 200 μL PBS buffer. The 60 μL bacterial suspension was added with 20 μL 4 × sample buffer (0.250 M Tris base, 0.28 M SDS, 40% glycerol, 20% 2-mercapto-ethanol, bromphenol blue), boiled at 100 °C for 10 min, and centrifuged at 12,000× *g* for 10 min. The supernatant was retained, and the target protein was resolved by 12% SDS-PAGE (SDS-PAGE Gel Kit, Solarbio, China). The gels were stained with InstaBlue protein stain solution (APE×BIO, Des Moines, IA, USA), and the results were observed with Gel Doc XR+ System (Bio-Rad, Hercules, CA, USA).

Proteins were electroblotted onto an NC membrane using semidry electrophoretic transfer cell (BIORAD, USA) following separation on 12% SDS-PAGE. The membrane was then blocked at room temperature for 3 h in blocking buffer (5% nonfat dried milk). The membrane was rinsed with PBST (1 × PBS, 1‰ tween 20) for 30 min after being treated with the anti-GST mouse monoclonal antibody (TransGen Biotech, Beijing, China) at 4 °C overnight. The membrane was then incubated with the goat anti-mouse IgG secondary antibody. Gel Doc XR+ System was used to analyze the data after staining the membrane with 2 mL of eECL chromogen (Cowin Bio., Beijing, China).

## 5. Conclusions

There are 23 chitinase genes in *F. vesca*, which can be divided into 5 subclasses. Most of the *FvChi* genes were differentially expressed in various organs of strawberry and showed upregulation following *C. gloeosporioides* inoculation. Further, some genes were strongly induced by hormone (SA and JA) treatment. In addition, overexpression of *FvChi-14* increased the resistance of *A. thaliana* to *C. higginsianum*, via regulating the SA and JA signaling pathways. The FvChi-14 protein may be localized in the cell wall or extracellular matrix. This study offers insight into the potential roles of strawberry chitinase genes in biotic resistance and provides new candidates for stress resistance breeding programs. The study provides a basis for the functional characterization of *FvChi* genes.

## Figures and Tables

**Figure 1 plants-12-01543-f001:**
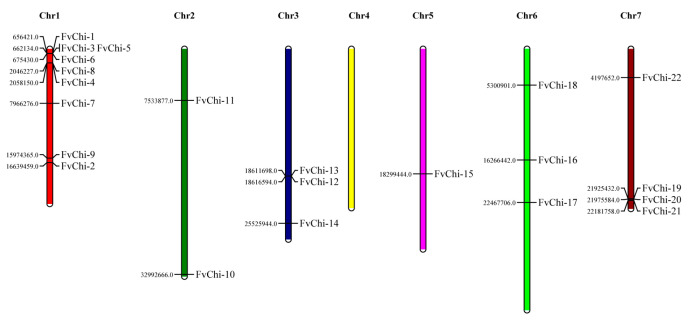
Chromosomal distribution of the *FvChi* genes. The numbers indicate where the genes start on the chromosomes. Based on their chromosomal order, the *FvChi* genes were given the names *FvChi-1* to *FvChi-23*.

**Figure 2 plants-12-01543-f002:**
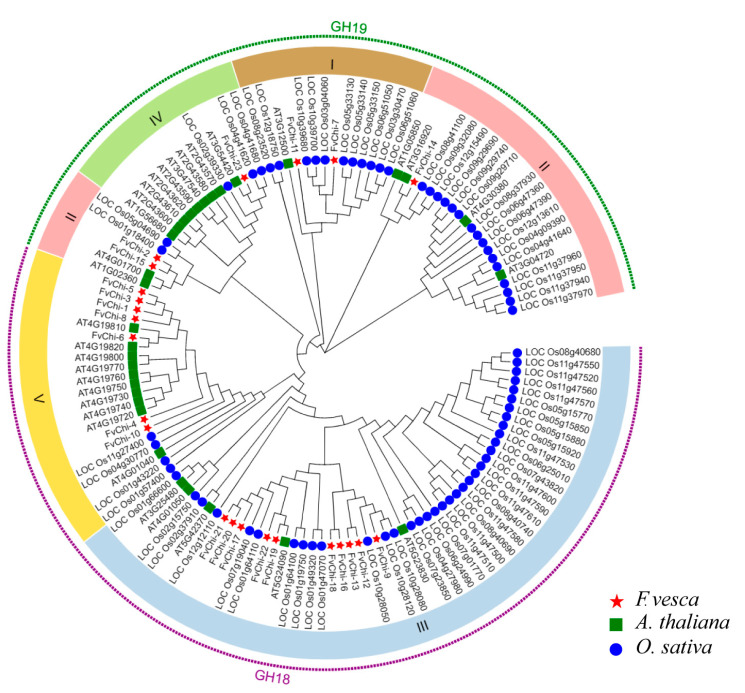
Phylogenetic analysis of chitinases proteins from *A. thaliana*, *O. sativa*, and *F. vesca.* Stars, squares, and circles represent *F. vesca*, *A. thaliana*, and *O. sativa* proteins, respectively.

**Figure 3 plants-12-01543-f003:**
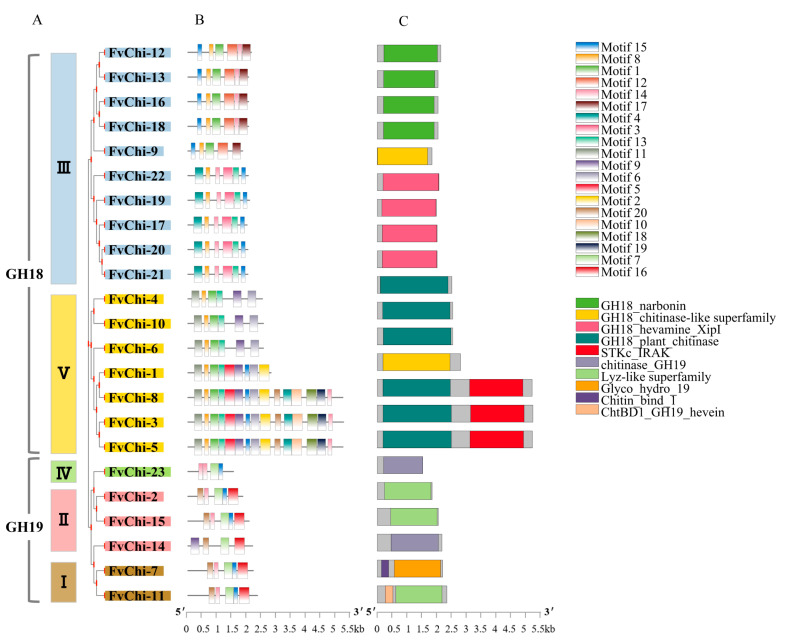
Gene structure analysis of *FvChi*. (**A**) Phylogenetic analysis and classification. Different boxes denote different subgroups. (**B**) Motif distribution patterns. (**C**) Domain distributions.

**Figure 4 plants-12-01543-f004:**
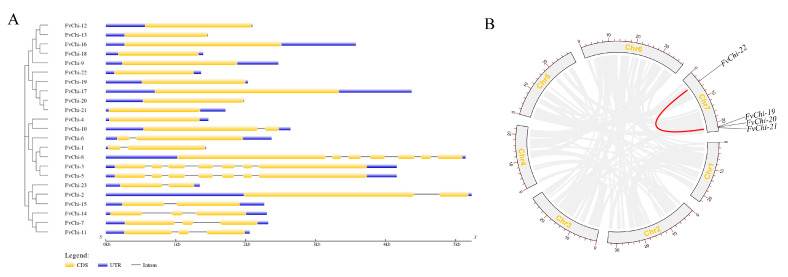
Exon–intron structure and collinearity analysis of *FvChi* genes. (**A**) Exon–intron structure of *FvChi* genes. UTRs are shown as blue boxes, exons as yellow boxes, and introns as black lines. (**B**) Collinearity analysis of *FvChi* genes. The syntenic relationships of *FvChi* genes are connected by a red line.

**Figure 5 plants-12-01543-f005:**
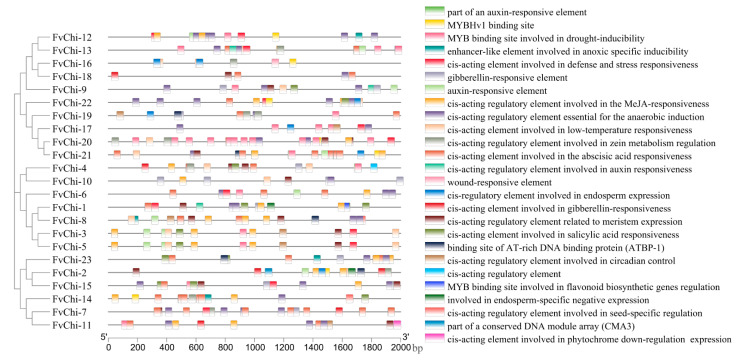
Promoter analysis of FvChi genes.

**Figure 6 plants-12-01543-f006:**
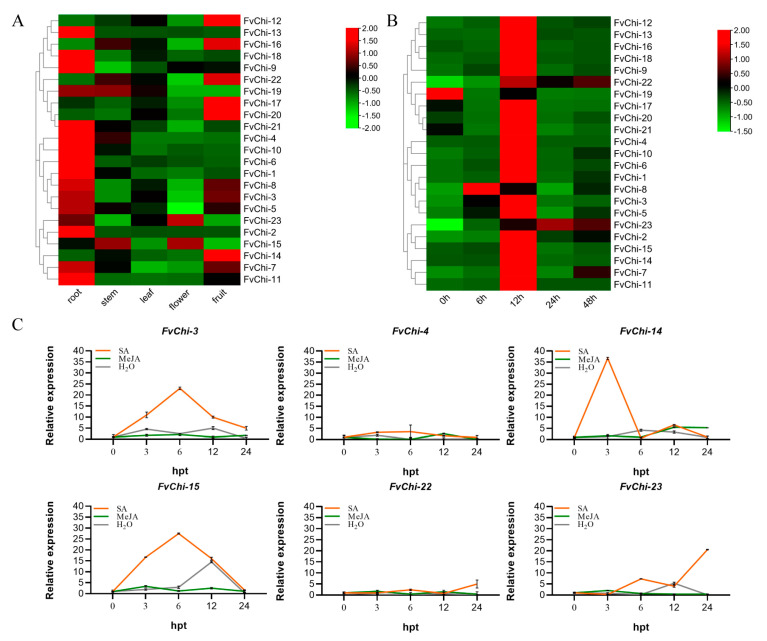
*FvChi* gene expression profiles in different tissue/organs and response to *C. gloeosporioides* inoculation and hormones treatment by qRT-PCR. (**A**) *FvChi* gene expression in different tissue/organs including root, stem, leaf, flower, and fruit. (**B**) *FvChi* gene expression profiles in response to *C. gloeosporioides* inoculation. (**C**) *FvChi* gene expression profiles in response to SA, MeJA, and water treatments. Red and green color scale denotes high and low expression levels, respectively. Error bars indicate the standard deviation (SD) of three biological replicates.

**Figure 7 plants-12-01543-f007:**
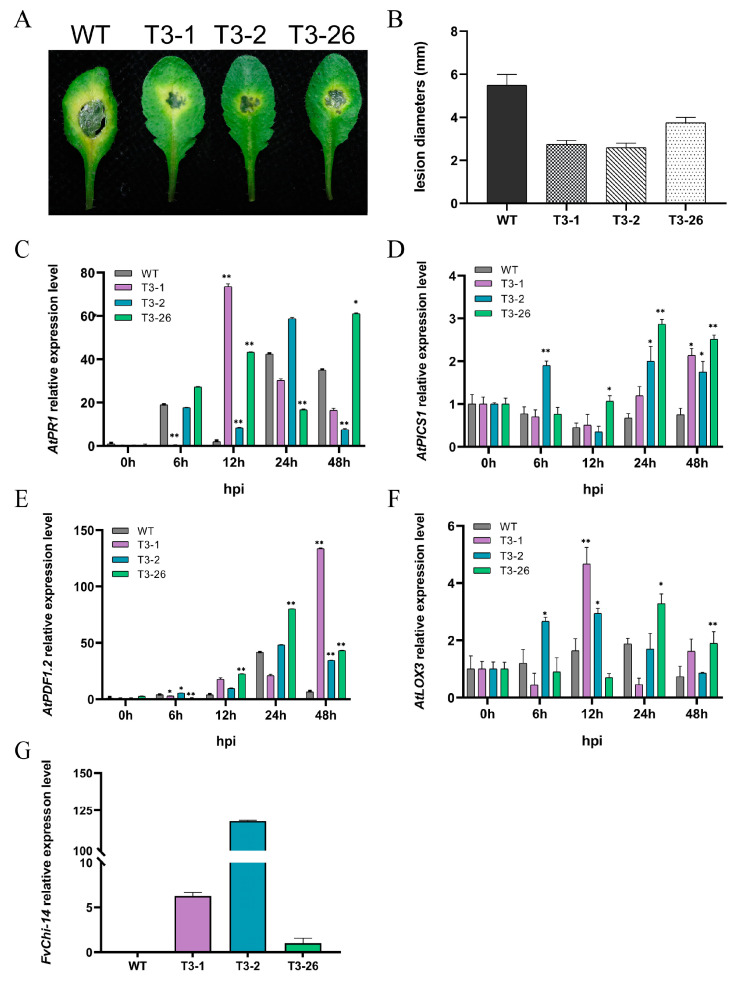
Overexpression of *FvChi-14* in *Arabidopsis* following *C. higginsianum* infection. Data are the mean values and SDs of three replications. (**A**) Symptoms of disease in Col-0 and transgenic *A. thaliana* leaves after 48 h of inoculation. (**B**) At 48 hpi, the average lesion diameter on leaves. (**C**) Expression analysis of *AtPR1* gene. (**D**) Expression analysis of *AtICS1* gene. (**E**) Expression analysis of *AtPDF1.2* gene. (**F**) Expression analysis of *AtLOX3* gene. (**G**) The transcript level of *FvChi-14* in leaves of WT and transgenic lines. Asterisks represent a statistically significant difference (* *p* < 0.05, ** *p* < 0.01, Student’s *t* test) between wild-type and transgenic lines.

**Figure 8 plants-12-01543-f008:**
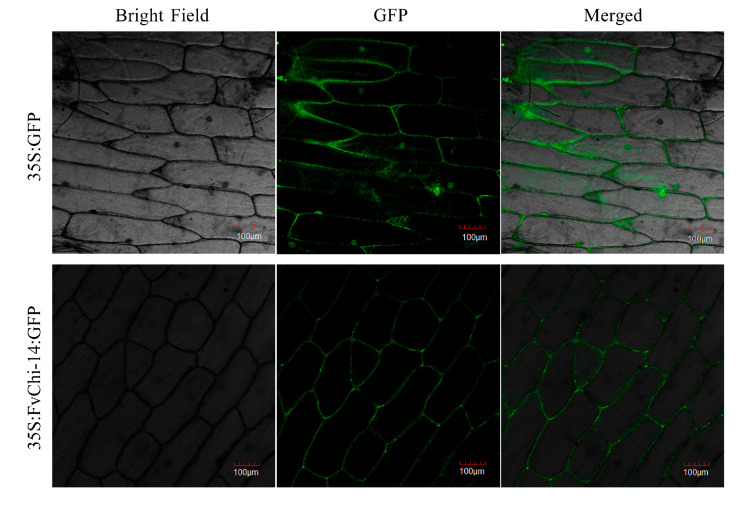
Subcellular localization of FvChi-14 in onion epidermal cells. Scale bar: 100 µm.

**Figure 9 plants-12-01543-f009:**
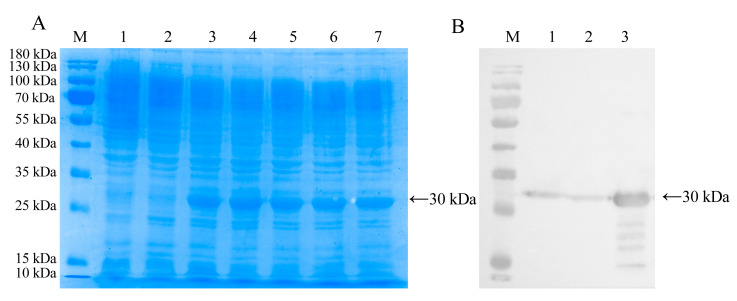
Prokaryotic expression of FvChi-14 in *E. coli* and immunoblotting analysis. Note: M: protein ladder; (**A**) 12% SDS-PAGE analysis of recombinant FvChi-14 protein. 1: pGEX-4T-1; 2: recombinant strain without IPTG induction; 3–7: recombinant strain induced by 0.1, 0.2, 0.4, 0.8, 1.6 mmol·L^−1^IPTG, respectively. (**B**) Immunoblotting analysis of the recombinant FvChi-14 protein with anti-GST antibody. 1: pGEX-4T-1; 2: recombinant strain without IPTG induction; 3: recombinant strain induced by 0.2 mmol·L^−1^IPTG.

**Table 1 plants-12-01543-t001:** The details of chitinase genes identified in *F. vesca*.

Name	Accession No.	Locus Name	Chr.	Location	CDS/bp	ORF/aa	Mw/kDa	pI	Predicted Location
*FvChi-1*	XM_004288652.2	XP_004288700.2	1	656,421–657,812	1248	415	46.867	8.91	cell wall
*FvChi-2*	XM_004288324.2	XP_004288372.1	1	16,639,459–16,642,667	825	274	30.213	8.72	extracell
*FvChi-3*	XM_004288653.2	XP_004288701.2	1	662,134–665,736	2334	777	87.535	7.59	chloroplast
*FvChi-4*	XM_011470234.1	XP_011468536.1	1	2,058,150–2,059,443	1119	372	41.070	5.91	cell wall
*FvChi-5*	XM_011461675.1	XP_011459977.1	1	662,134–665,736	2325	774	87.266	7.59	cell membrane
*FvChi-6*	XM_004287056.2	XP_004287104.1	1	675,430–677,228	1134	377	41.072	8.95	cell wall
*FvChi-7*	XM_004287696.2	XP_004287744.1	1	7,966,276–7,968,174	981	326	34.390	5.18	vacuole
*FvChi-8*	XM_011470322.1	XP_011468624.1	1	2,046,227–2,050,302	2322	773	86.727	6.4	cell membrane
*FvChi-9*	XM_004289370.2	XP_004289418.1	1	15,974,364–15,976,008	822	273	30.613	4.88	cell wall
*FvChi-10*	XM_004291897.2	XP_004291945.1	2	32,992,664–32,994,601	1134	377	41.423	8.9	cell wall
*FvChi-11*	XM_004289994.2	XP_004290042.1	2	7,533,877–7,535,599	1044	347	36.715	8.23	vacuole
*FvChi-12*	XM_004294591.2	XP_004294639.1	3	18,616,594–18,618,119	954	317	35.684	7.17	cell wall
*FvChi-13*	XM_004295915.2	XP_004295963.2	3	8,611,698–18,612,883	912	303	33.957	7.09	cell wall
*FvChi-14*	XM_004295058.2	XP_004295106.1	3	25,525,945–25,527,885	972	323	35.775	6.2	extracell
*FvChi-15*	XM_004299998.2	XP_004300046.2	5	18,299,444–18,301,125	918	305	33.816	8.56	vacuole
*FvChi-16*	XM_011468684.1	XP_011466986.1	6	16,266,442–16,268,685	912	303	33.529	4.7	cell wall
*FvChi-17*	XM_011468993.1	XP_011467295.1	6	22,467,706–22,470,336	888	295	31.370	4.64	vacuole
*FvChi-18*	XM_004304955.2	XP_004305003.1	6	5,300,901–5,302,053	912	303	33.551	5.05	cell wall
*FvChi-19*	XM_004307979.2	XP_004308027.1	7	21,925,431–21,926,910	927	308	33.000	8.62	vacuole
*FvChi-20*	XM_011471995.1	XP_011470297.1	7	21,975,584–21,977,021	900	299	31.611	7.51	vacuole
*FvChi-21*	XM_004308005.2	XP_004308053.1	7	22,181,758–22,183,063	900	299	31.562	5.28	vacuole
*FvChi-22*	XM_004306663.2	XP_004306711.2	7	4,197,652–4,198,786	906	301	32.382	8.08	vacuole
*FvChi-23*	XM_004309786.2	XP_004309834.1	/	355,466–356,526	684	227	24.662	4.59	extracell

Abbreviations: Chr.: chromosome; CDS: coding sequence; ORF: open reading frame; pI: isoelectric point.

## Data Availability

No new data were created or analyzed in this study. Data sharing is not applicable to this article.

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
