# Peer review of "Bioinformatics and Expression Analysis of the Chitinase Genes in Strawberry (Fragaria vesca) and Functional Study of FvChi-14"

_plants, 2023, doi:10.3390/plants12071543_

Round 1

Reviewer 1 Report

Well written and interesting article related to bioinformatic and expression analysis of the chitinase genes in strawberry. Research is well planned and performed, results suport conclusions. In my opinion some minor corrections should be done:

 Line 39- correct the word „trganisms”

Write more precisely why the FvChi14 gene was selected for further analysis.

 Describe more precisely RT-PCR experiments: how the quality of RNA was assessed, method of removing remnants of genomic DNA, amount of  RNA per analysed sample.

Author Response

Dear editors and reviewers,

Thank you for giving us the opportunity to submit a revised manuscript “Bioinformatics and Expression Analysis of the Chitinase Genes in Strawberry (Fragaria vesca) and Functional Study of FvChi-14” for publication in the Plants Journal.

We appreciate the time and effort that you and the reviewers dedicated to providing feedback on our manuscript and are grateful for the insightful comments on and valuable improvements to our paper. We have incorporated the suggestions made by the reviewers. Those changes are markered within the manuscript. Please see below, for a point-by-point response to the reviewer’s comments and concerns. We have corrected all the grammatical and contextual mistakes. We would be happy to consider any further changes that you deem appropriate and look forward to working with the editorial staff through the final submission process.

Thank you again, and best regards,

Dr. Zhifeng Wen

College of Horticulture,

Fujian Agriculture and Forestry University

Response to Reviewer 1 Comments

Point 1: Line 39- correct the word „trganisms”.

Response: Thank you for your suggestion and we apologize for this mistake. We have modified this word to ''organisms'' (Line 47).

Point 2: Write more precisely why the FvChi14 gene was selected for further analysis.

Response: Thank you for your comment. Bioinformatics and expression profiling revealed that the FvChi14 gene was abundantly induced in strawberries when inoculated with C. gloeosporioides, and the expression fold was maximally up-regulated with external hormone treatment (SA). Therefore, the FvChi14 gene was selected for more detailed analysis and overexpressed in Arabidopsis. (Line 218-221).

Point 3: Describe more precisely RT-PCR experiments: how the quality of RNA was assessed, method of removing remnants of genomic DNA, amount of  RNA per analyzed sample.

Response: Thanks to your suggestion, we have added details of qRT-PCR experiments (L 465-470).

Reviewer 2 Report

The manuscript describes the characterization of chitinase genes from strawberry plants and the functional analysis of FvChi-14. After identifying chitinase genes from the strawberry genome, the phylogenetic and gene structure analyses of chitinase genes are performed. Expression analysis finds the inducibility of chitinase genes after pathogen inoculation or plant hormone treatment. Overexpression of FvChi-14 in Arabidopsis enhances resistance against C. higginsianum. The function of FvChi-14 against osmotic stress was also speculated using recombinant E.coli.

The manuscript has novel information on chitinase genes and contributes to a better understanding of biotic and abiotic stress responses in the strawberry. However, several concerns should be addressed before publication, as listed below.

Major point

1) It is unclear whether organ-specif expression analysis (Figure 6A) is performed by qRT-PCR in this study or shows the transcriptome data from the previous report. Please clearly state this point in the main text and legend of Figure 6. The manuscript describes "total RNA extracted from treated strawberry leaves" in lines 414-415.

2) The reviewer agrees that Figure 7 shows the enhanced resistance of FvChi-14 overexpression lines (T3-1, T3-2, and T3-26). However, the manuscript describes that "three independent T3 transgenic lines (T3-1, T3-2, and T3-26) having the strongest resistance to C. higginsianum were chosen for further studies (lines 202-203)." Does this description mean the overexpression lines are not selected by the expression levels of FvChi-14? Are there lines not showing enhanced resistance despite the high-level expression of FvChi-14?

The overexpression lines should be selected by the level of FvChi-14 expression, not by the resistance to the pathogen, and then the inoculation test should be performed for the selected line. The expression levels of FvChi-14 in overexpression lines also should be shown.

3) Several marker gene expressions are not consistent in independent overexpression lines. For example, AtPR1 is upregulated in T3-26 at 48 hpi but down-regulated in T3-1 and T3-2. Therefore, the manuscript should carefully describe this inconsistency in the “results” and “discussions”.

 4) The antifungal activity of chitinase can explain the enhanced resistance overexpression lines of chitinase. If possible, the chitinase activity or antifungal activity of FvChi-14 is shown using recombinant protein.

5) Please discuss the possible mechanism of SA and JA signaling altered by chitinase overexpression in the manuscript. As mentioned 4, the antifungal activity of chitinase can explain the enhanced resistance overexpression lines. How does chitinase affect SA and JA signaling?

6) GFP also seems to localize the cell wall, plasma membrane, or extracellular matrix (Figure 8). Does GFP control have signal peptides? Does the plasmolysis of onion cells clarify the localization of GFP and GFP-FvChi14?

7) The quantitative data for E. coli viability or growth should be added (Figure 10). Why do cells expressing FvChi-14 grow better in LB medium, but the difference with the vector control disappears or becomes smaller under stress conditions? The discussion in lines 341-346 lacks this point. Is it possible to speculate that FvChi-14 has a negative effect on the growth of E. coli under stress? This experiment also has concerns, as listed below. The reviewer recommends that Figure 10 is excluded from the manuscript unless the issues are adequately addressed.

 - E.coli growth should be quantitatively analyzed by counting colony forming units (CFU) on a solid medium or growth curve in a liquid medium.

 - The test plate of E. coli growth under stress conditions does not contain IPTG (lines 476-478). Is the expression of FvChi-14 maintained on the test plate?

 - Acid and base conditions should be adjusted with appropriate buffers because the growth of E. coli may affect the pH of the medium.

Minor points

8) Please spell out “C. gloeosporioides” in the first appearance (line 16).

9) Please describe the concentration of primer (line 420).

Author Response

Dear editors and reviewers,

Thank you for giving us the opportunity to submit a revised manuscript “Bioinformatics and Expression Analysis of the Chitinase Genes in Strawberry (Fragaria vesca) and Functional Study of FvChi-14” for publication in the Plants Journal.

We appreciate the time and effort that you and the reviewers dedicated to providing feedback on our manuscript and are grateful for the insightful comments on and valuable improvements to our paper. We have incorporated the suggestions made by the reviewers. Those changes are markered within the manuscript. Please see below, for a point-by-point response to the reviewer’s comments and concerns. We have corrected all the grammatical and contextual mistakes. We would be happy to consider any further changes that you deem appropriate and look forward to working with the editorial staff through the final submission process.

Thank you again, and best regards,

Dr. Zhifeng Wen

College of Horticulture,

Fujian Agriculture and Forestry University

Response to Reviewer 2 Comments

Point 1: It is unclear whether organ-specif expression analysis (Figure 6A) is performed by qRT-PCR in this study or shows the transcriptome data from the previous report. Please clearly state this point in the main text and legend of Figure 6. The manuscript describes "total RNA extracted from treated strawberry leaves" in lines 414-415.

Response 1: Thank you for your comments. FvChi members showed tissue-specific expression patterns that were analyzed by qRT-PCR. We have added annotations in the main text page 7,“Expression patterns of FvChi members were analyzed by quantitative reverse transcription PCR (qRT-PCR) (Figure 6A)”( Figure 6, “by qRT-PCR”).

Point 2:The reviewer agrees that Figure 7 shows the enhanced resistance of FvChi-14 overexpression lines (T3-1, T3-2, and T3-26). However, the manuscript describes that "three independent T3 transgenic lines (T3-1, T3-2, and T3-26) having the strongest resistance to C. higginsianum were chosen for further studies (lines 202-203)." Does this description mean the overexpression lines are not selected by the expression levels of FvChi-14? Are there lines not showing enhanced resistance despite the high-level expression of FvChi-14?

The overexpression lines should be selected by the level of FvChi-14 expression, not by the resistance to the pathogen, and then the inoculation test should be performed for the selected line. The expression levels of FvChi-14 in overexpression lines also should be shown.

Response 2: We are grateful for the suggestion. We selected the transgenic lines showing the high expression levels of FvChi-14 and the strongest resistance to C. higginsianum. The expression levels of FvChi-14 in overexpression lines are shown (Figure 7G), and the description (Line 224-226).

Point 3: Several marker gene expressions are not consistent in independent overexpression lines. For example, AtPR1 is upregulated in T3-26 at 48 hpi but down-regulated in T3-1 and T3-2. Therefore, the manuscript should carefully describe this inconsistency in the “results” and “discussions”.

Response 3: We are grateful for the suggestion. The details of the differential expression levels have been added in "results" and "discussions"(Line 235-248,357-358).

Point 4: The antifungal activity of chitinase can explain the enhanced resistance overexpression lines of chitinase. If possible, the chitinase activity or antifungal activity of FvChi-14 is shown using recombinant protein.

Response 4: We thank the reviewers for pointing this out. Keeping in view the postulated hypothesis and designated objectives to be achieved, it is perhaps a different aspect that cannot be performed at the moment owing to technical glitches and time strains. However, the suggestion has been seriously noted to be incorporated in the upcoming study design and the answer will be executed as per line in the next works.

Point 5: Please discuss the possible mechanism of SA and JA signaling altered by chitinase overexpression in the manuscript. As mentioned 4, the antifungal activity of chitinase can explain the enhanced resistance overexpression lines. How does chitinase affect SA and JA signaling?

Response: Thanks for your suggestion. The description of the possible mechanism of chitinase regulation of SA and JA has been added (Line 367-382).

Point 6: GFP also seems to localize the cell wall, plasma membrane, or extracellular matrix (Figure 8). Does GFP control have signal peptides? Does the plasmolysis of onion cells clarify the localization of GFP and GFP-FvChi14?

Response: There was no signal peptide in the GFP control. We refer to the methods and analysis of onion cell subcellular gene localization in "Overexpression of Grapevine VvIAA18 Gene Enhanced Salt Tolerance in Tobacco" and" Molecular Cloning and Expression Analysis of a F-box Protein Gene FnFBOX1 and Its Promoter from Fragaria nilgerrensis". 35S:GFP was observed as a positive control with green fluorescence in all tissues of the cells. The fusion construct (FvChi14-GFP) exhibited green fluorescence as a possible subcellular localization of the target protein.

Point 7: The quantitative data for E. coli viability or growth should be added (Figure 10). Why do cells expressing FvChi-14 grow better in LB medium, but the difference with the vector control disappears or becomes smaller under stress conditions? The discussion in lines 341-346 lacks this point. Is it possible to speculate that FvChi-14 has a negative effect on the growth of E. coli under stress? This experiment also has concerns, as listed below. The reviewer recommends that Figure 10 is excluded from the manuscript unless the issues are adequately addressed.

 - E.coli growth should be quantitatively analyzed by counting colony forming units (CFU) on a solid medium or growth curve in a liquid medium.

 - The test plate of E. coli growth under stress conditions does not contain IPTG (lines 476-478). Is the expression of FvChi-14 maintained on the test plate?

 - Acid and base conditions should be adjusted with appropriate buffers because the growth of E. coli may affect the pH of the medium.

Response: Thanks for your suggestion. We added the experiment of E. coli growth curve, however, we find that in the liquid cultivation, there is no significant difference in the stress condition. We have deleted Figure 10 as per your suggestion.

Minor points

8) Please spell out “C. gloeosporioides” in the first appearance (line 16).

Response: Done as suggested (Line 17).

9) Please describe the concentration of primer (line 420).

Response: Done as suggested (Line 475).

Reviewer 3 Report

Dear Authors,

I have an opportunity to review manuscript entitled:” Bioinformatics and Expression Analysis of the Chitinase Genes in Strawberry (Fragaria vesca) and Functional Study of FvChi-14” in Plants MDPI Journal.

Authors concentrated on plant chitinases (EC 3.2.1.14) well-studied in many plant species, but not in strawberries (Fragaria vesca). Authors identified 23 FvChi genes from the genome of strawberry (F. vesca) and into GH18 and GH19 subfamilies based on phylogenetic relationships. Moreover, gene physicochemical properties, chromosomal location, exon-intron distribution, domain arrangement, and probe of localization in epidermal tissue were explored.

The introduction give the reader some clear background to the research, but in my opinion the role of chitinase in plant biotic stress were not enough underlined;

I understand that Authors compared some aspects in genes expression results with overexpressed Arabidopsis, but I could not understand why pathogen test was done in Arabidopsis plants instead of Fragaria,  especially that these pathogens (C. higginsianum, C. gloeosporioides and C. fragariae) infects strawberies as Authors stated in introduction part. I suggest to deeply explained it; Moreover, why the localization test was done on onion epidermal layer, not on strawberries or Arabidopsis;

I understand the reason of these research conducting, but the reader should known the clear aim of the study, so please, add it;

The materials and methods section is clearly written in a repetitive way;

Authors suggested that :” Tissue-specific expression was noticed in some genes and the rate of FvChi transcripts in roots was higher than in other parts of the plant (stolon, leaf, flower, and fruit)”, but unfortunately leaf, roots, flower, fruits are not a plant tissue. Furthermore, in materials and methods we can find “For tissue-specific expression, strawberry leaf, stolon, flower, fruit and root, samples were collected”- the same situation there is not tissue specific method;

Despite of the fact that some papers can be find use actin in plant -fungi interaction, please be aware because gene expression analyses was done according only one reference gene, and actin gene is not a good choice for comparison /stability in plant -pathogen interaction, in fungus also;

Figure 8 should be improved, because the microphotographs quality is not acceptable in current form;

Authors add the short summarizing conclusion and in discussion part we can find the statement: “These findings propose that chitinase genes play various and specific roles in regulating biological and abiotic stress in strawberry”, so please add the future prospects coming from obtained results to underline the importance of Author’s results;

Minor aspects:

-       - Please enlarge with good resolution figures 3, 4, 5 because the reader can easy lost the important data;

-       - I suggest to be aware and check if all genes described in manuscript are in italics;

Author Response

Dear editors and reviewers,

Thank you for giving us the opportunity to submit a revised manuscript “Bioinformatics and Expression Analysis of the Chitinase Genes in Strawberry (Fragaria vesca) and Functional Study of FvChi-14” for publication in the Plants Journal.

We appreciate the time and effort that you and the reviewers dedicated to providing feedback on our manuscript and are grateful for the insightful comments on and valuable improvements to our paper. We have incorporated the suggestions made by the reviewers. Those changes are markered within the manuscript. Please see below, for a point-by-point response to the reviewer’s comments and concerns. We have corrected all the grammatical and contextual mistakes. We would be happy to consider any further changes that you deem appropriate and look forward to working with the editorial staff through the final submission process.

Thank you again, and best regards,

Dr. Zhifeng Wen

College of Horticulture,

Fujian Agriculture and Forestry University

Response to Reviewer 3 Comments

We have corrected all the grammatical and contextual mistakes.

Point 1: The introduction give the reader some clear background to the research, but in my opinion the role of chitinase in plant biotic stress were not enough underlined.

Response: Thanks for your suggestion. We have added more information in the introduction part (L 33-35, L 39-43).

Point 2: I understand that Authors compared some aspects in genes expression results with overexpressed Arabidopsis, but I could not understand why pathogen test was done in Arabidopsis plants instead of Fragaria, especially that these pathogens (C. higginsianum, C. gloeosporioides and C. fragariae) infects strawberries as Authors stated in introduction part. I suggest to deeply explained it; Moreover, why the localization test was done on onion epidermal layer, not on strawberries or Arabidopsis.

Response: Thanks for your suggestion, since overexpression in Fragaria needs more time and resources. Keeping in view the postulated hypothesis and designated objectives to be achieved, we selected expressing genes in Arabidopsis. Overexpression in Fragaria cannot be performed at the moment owing to technical glitches and time strains. However, the suggestion has been seriously noted to be incorporated in the upcoming study design and the answer will be executed as per line in the next works. The onion epidermis is the most commonly used transient expression system. Onion epidermis cells are large and have a typical plant cell structure.

Point 3: I understand the reason of these research conducting, but the reader should known the clear aim of the study, so please, add it.

Response: Thanks for your suggestion, Added as suggested (Line 93-96).

Point 4: The materials and methods section is clearly written in a repetitive way.

Response: Thanks for your suggestion. Changes have been made as suggested.

Point 5: Authors suggested that :” Tissue-specific expression was noticed in some genes and the rate of FvChi transcripts in roots was higher than in other parts of the plant (stolon, leaf, flower, and fruit)”, but unfortunately leaf, roots, flower, fruits are not a plant tissue. Furthermore, in materials and methods we can find “For tissue-specific expression, strawberry leaf, stolon, flower, fruit and root, samples were collected”- the same situation there is not tissue specific method.

Response 5: Thanks for your comments, We have revised the lines as “Organ/Tissue-specific expression was noticed in some genes and the rate of FvChi transcripts in roots was higher than in other parts of the plant (stolon, leaf, flower, and fruit)”(Line 478-481).

Point 6: Despite of the fact that some papers can be find use actin in plant -fungi interaction, please be aware because gene expression analyses was done according only one reference gene, and actin gene is not a good choice for comparison /stability in plant -pathogen interaction, in fungus also.

Response: We appreciate the suggestion, and we will pay attention to the selection of the reference genes in the subsequent study.

Point 7 : Figure 8 should be improved, because the microphotographs quality is not acceptable in current form.

Response: Thanks for the suggestion. A new figure having good resolution has been added(Line 264).

Point 8 : Authors add the short summarizing conclusion and in discussion part we can find the statement: “These findings propose that chitinase genes play various and specific roles in regulating biological and abiotic stress in strawberry”, so please add the future prospects coming from obtained results to underline the importance of Author’s results.

Response: Thank you for your comments we added the future prospects at lines 544-545.

Point 9 : Please enlarge with good resolution figures 3, 4, 5 because the reader can easy lost the important data.

Response: Thanks for pointing it out. Done as suggested.

Point 10 : I suggest to be aware and check if all genes described in manuscript are in italics.

Response: We checked the gene names and all are italicized.

Round 2

Reviewer 2 Report

The revised manuscript has addressed the issues raised in a previous review.

Reviewer 3 Report

Authors improved most of suggested points - Some of them [like more than one refrerence gene are not improved] I understand difficulties, despite of it I do not have an intension to block the mnauscript;

Presenting of figures, material and methods section information as well as introduction and aim of studies are significantly improved.